# rhEGF Treatment Improves EGFR Inhibitor-Induced Skin Barrier and Immune Defects

**DOI:** 10.3390/cancers12113120

**Published:** 2020-10-25

**Authors:** Ji Min Kim, Jun Ho Ji, Young Saing Kim, Suee Lee, Sung Yong Oh, Seok Jae Huh, Choon Hee Son, Jung Hun Kang, So Yun Ahn, Jung Eun Choo, Ki-Hoon Song, Mee Sook Roh

**Affiliations:** 1Bio Process Development Team, Life Science Research Institute, Daewoong Pharmaceutical Co., Ltd., Yongin 17028, Korea; jmkim102@daewoong.co.kr (J.M.K.); syahn090@daewoong.co.kr (S.Y.A.); jec1113@daewoong.co.kr (J.E.C.); 2Department of Internal Medicine, Samsung Changwon Hospital, Sungkyunkwan University School of Medicine, Changwon 51353, Korea; jijunho@skku.edu; 3Department of Internal Medicine, Gachon University Gil Medical Center, Incheon 21565, Korea; zoomboom@gilhospital.com; 4Department of Internal Medicine, Dong-A University Hospital, Busan 49201, Korea; sueelee@dau.ac.kr (S.L.); doctorhsj@dau.ac.kr (S.J.H.); 5Department of Pulmonology, Dong-A University Hospital, Busan 49201, Korea; chshon@dau.ac.kr; 6Department of Internal Medicine, Gyeongsang National University Hospital, Jinju 52727, Korea; newatp@gnu.ac.kr; 7National Cancer Center, Department of Dermatology, Goyang 10408, Korea; khsong@ncc.re.kr; 8Department of Pathology, Dong-A University College of Medicine, Busan 49201, Korea

**Keywords:** epidermal growth factor, epidermal growth factor inhibitor related skin rash, cetuximab, gefitinib

## Abstract

**Simple Summary:**

In our prior study, we demonstrated that recombinant human epidermal growth factor (rhEGF) treatment is effective for managing epidermal growth factor receptor inhibitors (EGFRIs)-related skin toxicities and improves patients’ quality of life (QoL) compared with placebo. Nevertheless, the mechanisms of rhEGF effects are unknown yet so basic study is needed to clarify the mechanisms. In this study, we revealed that treatment of rhEGF in human epidermal keratinocytes, 3d-cultured human skin tissue and patient lesions improved EGFRIs-induced skin eruption via normalizing proliferation and differentiation of keratinocytes, reducing inflammatory cytokines expression and inducing expression of AMPs. These findings provided an evidence for the use of rhEGF as a treatment for skin side effects derived from EGFRI.

**Abstract:**

The mechanisms of epidermal growth factor (EGF) affecting EGF receptor inhibitor (EGFRI)-related skin toxicities are as yet unknown. We investigated which mechanisms are involved in EGF’s positive effects. Two types of EGFRIs, cetuximab and gefitinib, were used to treat the cells or 3d-cultured human skin tissue with recombinant human EGF (rhEGF). As a result, rhEGF increased EGFR and pEGFR expression. Furthermore, rhEGF induces EGFR signaling by pAKT and pPI3K expression in gefitinib and rhEGF co-treated cells. In addition, rhEGF bound to EGFR after than cetuximab, but cetuximab bound to EGFR more strongly than rhEGF. Moreover, expressions of proliferation and differentiation proteins, both ki-67 and filaggrin, were decreased in EGFRI-treated tissue. However, in rhEGF and EGFRI co-treated tissue, those expressions were increased. Expression of IL-1α, IL-8, and TNF-α was increased by EGFRIs and down-regulated by rhEGF. Furthermore, hBD-2 and hBD-3 protein expressions were inhibited by cetuximab or gefitinib treatment, and those decrements were increased by rhEGF treatment. In patients’ tissue evaluation, compared with controls, patients’ Ki-67 and EGFR expression were decreased (*p* = 0.015, *p* = 0.001). Patients’ IL-17 and TNF-α expression intensity was higher than that of the control group (*p* = 0.038, *p* = 0.037). After treatment with EGF ointment, average values of Ki-67, EGFR, and Melan-A were changed to normal values. Oppositely, patients’ proportions of IL-17 and TNF-α were decreased to low stain level. In conclusion, treatment of rhEGF improved EGFRI-induced skin eruption via normalizing the proliferation and differentiation of keratinocytes, reducing inflammatory cytokines by the affected EGFRIs.

## 1. Introduction

Epidermal growth factor receptor (EGFR) inhibition is a good target for the treatment of diverse metastatic epithelial cancers, including lung, colon, pancreatic, and head and neck cancers [1,2,3]. There are two strategies to inhibit EGFR signaling: monoclonal antibodies and tyrosine kinase inhibitors (TKI). Despite the treatment approach, use of EGFR inhibitors (EGFRIs) is associated with adverse side effects [4]. Among the cutaneous toxicities observed in cancer patients treated with EGFRIs are papulopustular rash of the upper trunk and face skin in (60–90%) and dry and itchy skin in (12–16%) [5,6]. Although the side effects induced by EGFRIs are generally classified as moderate, they are usually chronically persistent, may significantly impact the patient’s quality of life (QoL), and thus necessitate dose reduction, or even interruption of treatment.

The most frequently used formulation for EGFRI-induced skin reactions is the oral or topical application of antibiotics, such as the tetracycline family and corticosteroids, alone or in combination with moisturizers and sunscreen [7,8,9,10,11,12]. Antibiotics, including doxycycline and minocycline, are commonly used to treat acneiform rash and help reduce its symptoms. Topical treatment of corticosteroids is also generally used for treatment of EGFRI-related skin adverse events (ERSEs), especially for skin rash. Other topical agents, such as retinoid and vitamin K1 cream, have potential roles for the management of ERSEs [13,14]. However, their efficacies have not been fully investigated through prospective studies. A recent phase III trial has shown that the prophylactic use of vitamin K1 cream in combination with doxycycline cannot decrease the incidence of grade ≥ 2 skin rash in patients initiating cetuximab therapy, compared to doxycycline and vehicle [15]. Even though these therapeutics are used to treat ERSE, they could not treat the main reason, but only relieve the symptoms.

The epidermal growth factor (EGF) plays a key role in wound healing, epidermal keratinocytes are a rich source of EGFR ligands, and EGFR signaling has a major effect on the proliferation and differentiation of keratinocytes [16]. Therefore, EGF plays an important role in skin development and homeostasis [17]. Beyond its role in wound healing and epithelial homeostasis, recent studies have revealed that EGF has a protective effect of skin barrier functions in atopic dermatitis and acne vulgaris [17,18,19,20]. Hershey et al. reported that EGF had an immunomodulatory role in inflamed skin tissue, showing that EGFR signaling reduces allergen-induced IL-6 production and Th17 responses in the skin [17]. Similarly, our previous studies revealed that EGF treatment regulates TLR-2-induced inflammatory reaction in human epidermal keratinocytes [20], and topical treatment of EGF relieved S. aureus-induced inflammation and AD-like skin lesions in Nc/Nga mice [18].

In a previous study, we demonstrated that EGF ointment is effective at managing EGFR inhibitor-related skin toxicities and improves patients’ QoL, compared with placebo, via a placebo-controlled, double-blind, multicenter, pilot phase III trial [21]. Nevertheless, the mechanisms of EGF effects are as yet unknown, so basic study is needed to clarify the mechanisms. In this study, we therefore investigated which mechanisms are involved in EGF’s positive effects on EGFRI-induced skin irruptions using human epidermal keratinocytes and 3D-cultured human skin tissue. In addition, we compared in vitro and ex vivo results with the tissues of patients who treated EGF topically.

## 2. Materials and Methods

### 2.1. Reagents

For the purposes of this study, recombinant human EGF (rhEGF, Daewoong Pharmaceutical Co., Ltd., Seoul, Korea) was used. Cetuximab (5 mg/mL, Erbitux^®^) was purchased from Merck (Darmstadt, Germany), and gefitinib (250 mg, Iressa^®^) was purchased from AstraZeneca Corporation (San Diego, CA, USA). To induce inflammatory reaction, lipopolysaccharide (LPS, Sigma-Aldrich, St. Louis, MI, USA) was used, and it was treated for 24 h before EGFRIs or rhEGF treatment.

### 2.2. Cell Culture

Primary human epidermal keratinocytes (Thermo Fisher Scientific, Waltham, MA, USA) were cultured in EpiLife Medium (Thermo Fisher Scientific) with Human Keratinocyte Growth Supplement (HKGS, Thermo Fisher Scientific). The cells were maintained in a humidified atmosphere of 5 % CO_2_ and 37 °C, and the medium was replaced every two days. Before reagent treatment, the cells were cultured overnight in EpiLife Medium without HKGS to induce starvation. To induce inflammatory reaction, 100 ng/mL of LPS was treated for 24 h. The concentration of LPS was decided by several references [22,23,24] Then, various concentrations of rhEGF with EGFRIs (10 ng/mL cetuximab or 1 μM gefitinib) or only EGFRIs were treated to LPS-treated keratinocytes for (24 or 48 h).

### 2.3. 3D-Cultured Human Skin Tissue

Neoderm^®^ is a 3D human skin tissue model in which human primary keratinocytes and fibroblasts are 3-dimensionally cultured to mimic the morphology and physiology of human skin. To confirm whether rhEGF affects epidermis homeostasis, the Neoderm^®^ tissues were cultured with rhEGF and EGFRI (10 ng/mL cetuximab or 1 μM gefitinib) co-treatment or only EGFRI treatment for 48 h. Moreover, to investigate whether rhEGF affects inflammatory reaction, the Neoderm^®^ tissues were cultured with LPS for 24 h. After that, EGFRIs (10 ng/mL cetuximab or 1 μM gefitinib) and/or 10 ng/mL EGF were treated to LPS-treated Neoderm^®^ tissues for 48 h. The reagents were treated to the top of tissues. Then, each tissue was fixed by a 4% formaldehyde solution.

### 2.4. Real-Time Quantitative PCR

After reagents treatment for 24 h, the cells were harvested by trypsinization, the total RNA was extracted using the ReliaPrep^™^ RNA Cell Miniprep System (Promega, Madison, WI, USA), and 1 μg of the total RNA was converted to cDNA using the High-Capacity RNA-to-cDNA^™^ Kit (Applied Biosystems^™^, Foster City, CA, USA), under the following reaction conditions: 45 °C for 45 min, and 95 °C for 5 min. Probes were obtained from Applied Biosystems as Assays-on-Demand^™^ Gene Expression Assays (glyceraldehyde-3-phosphate dehydrogenase [GAPDH]: Hs02758991_g1, IL-1α: Hs00174092_m1, IL-8: Hs00174103_m1, TNF-α: Hs01113624_m1, IL-17α: Hs00174383_m1, IL-17β: Hs07287652_m1, hBD-1: Hs00608345_m1, hBD-2: Hs00175474_m1, hBD-3: Hs04195435_g1, hBD-4: Hs00414476_m1, LL37: Hs00189038_m1, RNase 7: Hs00922963_s1). Reactions were carried out on the ABI StepOnePlus^™^ (Applied Biosystems), and relative transcription levels were determined by GAPDH as the reference gene. The data were analyzed using the ABI StepOnePlus^™^ software (Applied Biosystems).

### 2.5. Western Blot Assay

To confirm whether rhEGF activated EGFR on human keratinocytes when rhEGF and cetuximab were co-treated to cells, EGFR and phosphorylated EGFR (pEGFR) expressions in human keratinocytes were detected by the Western blot assay. Various concentrations of rhEGF and 10 ng/mL cetuximab were co-treated to cells for 4 h, and then the cells were washed by PBS. Then, the cell pellet was harvested and stored at −80 °C in a deep freezer, before use.

To confirm whether rhEGF activated EGFR on human keratinocytes when rhEGF and gefitinib were co-treated to cells, EGFR signaling markers, including AKT, phosphorated AKT (pAKT), PI3K (pPI3K), and phosphorylated PI3K, were detected by the Western blot assay. Various concentrations of rhEGF and 1 μM gefitinib were co-treated to cells for 40 min, and the cells were washed by PBS 1 time, harvested, and stored at −80 °C in a deep freezer, before use.

To detect the expression of EGFR, pEGFR, AKT, pAKT, PI3K, and pPI3K in reagents-treated keratinocytes, the cells were lysed in a RIPA lysis and extraction buffer (Thermo Fisher Scientific, Waltham, MA, USA), to which was added a protease inhibitor cocktail (Thermo Fisher Scientific), and the total protein concentration was measured using the BCA assay (Thermo Fisher Scientific). The 30 g of total protein per sample was resolved using SDS-PAGE on a 4–12% Bis-Tris gel (Nupage; Invitrogen Corp., Carlsbad, CA, USA), using MES SDS running buffer (Nupage, Invitrogen). Then, the resolved protein was transferred to a PVDF membrane using iBlot gel transfer device (Thermo Fisher Scientific). Western blot analysis was performed according to standard procedures, using each primary antibody. The reaction product was detected by enhanced chemiluminescence (Amersham Imager 600, GE Healthcare Life Sciences, IL, USA). All primary antibodies including EGFR, pEGFR, AKT, pAKT, PI3K, pPI3K, and β-actin were purchased at Cell Signaling Technology (Danvers, MA, USA). The antibodies were diluted according to the manual of each primary antibody. The original Western blot figures can be found in the Appendix A.

### 2.6. Receptor Binding Affinity Test

The binding affinity of rhEGF and cetuximab to human EGFR (R&D Systems, Minneapolis, MN, USA) was measured by Biacore T200 (GE Healthcare, Chicago, IL, USA). HBS-EP was used as running buffer, and 30 mM NaOH was used for regeneration of the chip surface. The concentrations were (1.563, 3.125, 6.25, 12.5, 25, 50, 100, and 200) nM for rhEGF and (0.012, 0.024, 0.049, 0.098, 0.195, 0.391, 0.781, 1.563, 3.125, 6.25, 12.5, 25, 50, 100, and 200) nM for cetuximab. Because of the difference of molecular weight between rhEGF and cetuximab, rhEGF was analyzed by 1:1 model, while cetuximab was analyzed by bivalent model. The association constant (Ka) and the equilibrium dissociation constant (KD) were obtained to evaluate the binding affinity by using BIA evaluation software version 3.0 (GE Healthcare).

### 2.7. Tissue Pathology

Twelve patients’ ERSEs skin biopsies from randomized clinical trials of EGF treatment for ERSE were obtained (clinicaltrials.gov NCT02284139) [21]. The same numbers of comparison-controlled skin biopsies who did not undergo a related clinical trial were obtained from the Bio-Resource Bank and Dong-A University Hospital. Among them, five patients were treated with EGF ointment for ERSE. We also observed skin changes before and after EGF ointment.

#### 2.7.1. Hematoxylin and Eosin

All tissues, including 3D-cultured human tissues and biopsy samples of patients, were fixed by 4% formaldehyde solutions. Then, the fixed tissues were embedded in paraffin. The 4 μM tissue sections were deparaffinized and rehydrated in a graded ethanol series. Then, sections were stained with hematoxylin-eosin (H&E). All slides were examined under light-microscopy (Olympus, Tokyo, Japan).

#### 2.7.2. Immunohistochemical (IHC) Assay

Immunohistochemistry was performed on formalin-fixed paraffin-embedded sections, according to the manufacturers’ instructions. The panel of primary antibodies included: Ki-67 antibody (Dako, Zug, Switzerland); EGFR Pharm Dx Kit (Agilent, Glostrup, Denmark); Melan-A antibody (Fuzhou Maixin Biotechnology Co. Ltd., Fuzhou, China); IL-17 (dilution 1:200; Abcam Inc., Cambridge, UK); and TNF-α antibody (MyBioSource, San Diego, CA, USA). Immunohistochemistry was performed using a Ventana BenchMark automated stainer (Ventana Medical Systems, Tucson, AZ, USA). The immunohistochemistry assessments were evaluated by a pathologist. Each specimen was analyzed three times for each different field at ×200 magnification. We performed the human investigations after approval by Dong-A University Hospital’s institutional review board (DAUH-IRB-19-003).

#### 2.7.3. Immunofluorescence (IF) Assay

The paraffin-embedded 3D-cultured human tissue sections were deparaffinized with xylene, dehydrated in gradually decreasing concentrations of ethanol, and then subsequently treated with 3% hydrogen peroxidase in TBS for 30 min to block endogenous peroxidase activity. Then, antigen retrieval was performed by using low pH IHC antigen retrieval solution (Invitrogen) at 90 °C for 30 min. The sections were immediately immersed in TBS for 10 min at 4 °C. Thereafter, the sections were blocked with BlockAid^™^ Blocking Solution (Invitrogen) for 15 min at room temperature (RT). The sections were rinsed in PBS and incubated with primary antibody overnight at 4 °C, and a secondary antibody was performed for 2 h at RT. The nuclei stain and mounting were performed with ProLong^™^ Gold Antifade Mountant with DAPI (Invitrogen). Appendix A lists the Primary antibodies and goat anti-rabbit or goat anti-mouse secondary antibody with Alexa Fluor 488 (Invitrogen) that were used in this study.

### 2.8. Statistical Analysis

All experiments were carried out in triplicate, and the results are expressed as mean ± standard deviation. 𝑃 values < 0.05 were considered statistically significant. One-way analysis of variance with Dunnett’s post-test was performed using GraphPad Prism version 8 (GraphPad Software, San Diego, CA, USA).

## 3. Results

### 3.1. rhEGF Activated EGFR Signaling When EGFRIs and rhEGF Co-Treated to Keratinocytes

To investigate whether rhEGF could activate EGFR on keratinocytes when rhEGF and EGFRIs were co-treated to cells, we examined the phosphorylated EGFR expression in keratinocytes that were treated with rhEGF and cetuximab. As a result, when various concentration of rhEGF and cetuximab were co-treated to keratinocytes, the expressions of both EGFR and phosphorylated EGFR (pEGFR) were increased (Figure 1a). In particular, the highest expression was in pEGFR in 10 ng/mL rhEGF- and cetuximab-treated cells (Figure 1a). Then, to investigate whether EGFR could be activated by rhEGF when rhEGF was co-treated with gefitinib, the expression of EGFR signaling molecules, including AKT, PI3K, phosphorylated AKT (pAKT), and phosphorylated PI3K (pPI3K), in rhEGF and cetuximab co-treated keratinocytes was detected by a Western blot assay. Figure 1b shows that in rhEGF and gefitinib co-treated cells, pAKT and pPI3K were detected, and the expression of these molecules was increased by rhEGFR in a concentration-dependent manner (Figure 1b).

Next, we compared the binding affinity between rhEGF and cetuximab using BIACORE T200 (GE Healthcare). The value of each association constant (K_A_) was 6.07 × 10^5^ (rhEGF) and 3.54 × 10^5^ (cetuximab) (Table 1). This result meant that rhEGF bound to EGFR faster than cetuximab. However, in the results of the equilibrium dissociation constant (K_D_), the value of K_D_ of rhEGF (1.30 × 10^−9^) was lower than that of cetuximab (0.49 × 10^−9^), and this result meant that cetuximab bound to EGFR more strongly than rhEGF (Table 1).

### 3.2. rhEGF Had Regulatory Effects on the Proliferation and Differentiation of Keratinocytes in Ex Vivo Studies

To elucidate whether the proliferation/differentiation of keratinocytes altered by EGFRIs was regulated by rhEGF, we analyzed the expression of proliferation and differentiation markers including ki67, filaggrin, K5, and K10 in 3D-cultured human skin tissues using an immunofluorescence assay. Figure 2a shows that, compared with un-treated controls, ki67 expression in tissues treated with 10 ng/mL cetuximab or 1 μM gefitinib was decreased (Figure 2a). However, in 10 ng/mL rhEGF- and EGFRIs-treated tissues, ki67 expression was similar to ki67 expression in controls (Figure 2a). Similarly, filaggrin expression was decreased in cetuximab- or gefitinib-treated tissues, compared with untreated controls (Figure 2b). However, in rhEGF and cetuximab or gefitinib co-treated tissues, filaggrin expression was increased, compared with that of tissues treated only with cetuximab or gefitinib (Figure 2b).

Figure 2c shows that K5 expression was not affected by cetuximab treatment or rhEGF and cetuximab co-treatment. However, in tissue treated with gefitinib-only, K5 expression was down-regulated (Figure 2c). Notably, in rhEGF and gefitinib co-treated tissue, K5 expression level was similar to the expression of K5 in untreated controls (Figure 2c). Interestingly, K10 expression was increased by cetuximab or gefitinib treatment (Figure 2d), and that increment was not observed in rhEGF and cetuximab or gefitinib co-treated tissue (Figure 2d). Nevertheless, K10 expression level in rhEGF and cetuximab or gefitinib co-treated tissue was a little higher than in untreated controls (Figure 2d).

### 3.3. rhEGF Had Little Effect on the Expression of Tight Junction Proteins of Keratinocytes Including Claudin-1, -3, and Occludin

The effect of EGFRIs and rhEGF on the proliferation and differentiation of keratinocytes was observed. Thus, we evaluated the expression of tight junction proteins, including claudin-1, -3, and occludin, using an IF assay to investigate whether rhEGF and EGFRIs can affect tight junction proteins. Figure 3a shows that the reduction of claudin-1 expression was observed in EGFRIs-treated, and EGFRIs-only or rhEGF co-treated tissue. Interestingly, there was a difference between cetuximab-treated tissue and gefitinib-treated tissue. In cetuximab-treated tissue, claudin-1 decrement was observed in all epidermal layer, but in gefitinib-treated tissue, claudin-1 expression was decreased above the basal layer (Figure 3a).

Then, we observed claudin-3 expression in reagent-treated or non-treated 3D-cultured tissues. As a result, claudin-3 expression was observed in the stratum granulosum layer of untreated tissues (Figure 3b). In cetuximab- or gefitinib-treated tissue, claudin-3 expression was not observed (Figure 3b); and in cetuximab and rhEGF co-treated tissue, claudin-3 expression was also not detected (Figure 3b). However, in gefitinb- and rhEGF-treated tissue, claudin-3 expression was observed at the stratum granulosum layer (Figure 3b).

Next, we investigated the change of occludin expression between reagent untreated controls and reagent-treated tissues. Figure 3c shows that occludin was detected in all the epidermal layers of untreated or reagent-treated tissues. Only in EGFRI-treated tissues was it observed that occludin expression was slightly decreased (Figure 3c). Moreover, in rhEGF and EGFRIs co-treated tissues, occludin expression was slightly decreased, compared with untreated control, but it did not differ, compared only with EGFRI-treated tissues.

### 3.4. rhEGF-Inhibited Expression of Pro-Inflammatory Cytokines, including IL-1α, IL-8, and TNF-α, in In Vitro and Ex Vivo Studies

Next, we examined the effects of EGFRIs and rhEGF on inflammatory cytokine production from primary human epidermal keratinocytes and 3D-cultured human skin tissues that were stimulated by lipopolysaccharides. mRNA expression of IL-1α, IL-8, and TNF-α was increased by cetuximab or gefitinib treatment in LPS-pretreated keratinocytes (Figure 4a,b). Hence, in rhEGF and EGFRI co-treated cells, the increment of mRNA expression of cytokines was inhibited in an rhEGF concentration-dependent manner (Figure 4a,b). Similarly, as a result, the protein expression of cytokines was increased by EGFRI treatment in LPS-pretreated keratinocytes (Figure 4c,d). As a result, the protein expression of IL-1α, IL-8, and TNF-α in (1 and 10) ng/mL rhEGF and cetuximab or gefitinib co-treated keratinocytes (Figure 4c,d) was decreased. However, in 20 ng/mL rhEGF and EGFRIs co-treated cells, the protein expression was not decreased.

Then, to confirm the change of protein expression by rhEGF and EGFRI treatment, we assessed cytokines’ expression in EGFR or rhEGF and EGFR co-treated 3D-cultured tissues that were pre-treated with LPS. Figure 4e shows that IL-1α was detected in LPS-treated tissue, and cetuximab and gefitinib treatment increased IL-1α expression in the basal layer and stratum corneum of LPS-treated tissues. As in previous in vitro results, in rhEGF and EGFRIs co-treated tissues, IL-1α expression was reduced, but it was higher than in untreated controls.

The results of IL-8 and TNF-α expression were similar to that of IL-1a. Figure 4f shows that IL-8 expression in LPS-treated tissue was increased at the upper basal layer. It was observed that IL-8 expression in cetuximab- or gefitinib-treated tissue was higher than that of LPS-treated tissue. Interestingly, IL-8 expression in cetuximab-treated tissues was higher than that of gefitinib-treated tissues. Moreover, it was observed that rhEGF and EGFRI treatment decreased IL-8 expression. Similarly, TNF-α expression was higher in EGFRI-treated tissues than in LPS-treated tissue (Figure 4g). In cetuximab-treated tissue, TNF-α was detected in the basal layer and stratum corneum, but TNF-α was detected in all epidermal layers of the gefitinib-treated tissue. Figure 4g shows that rhEGF and EGFRI co-treatment decreased TNF-α expression.

### 3.5. rhEGF Increased the Antimicrobial Peptides Expression in Human Epidermal Keratinocytes, Especially HBD-2 and -3

The regulatory effect of rhEGF on EGFRIs-induced inflammatory cytokine expression was observed. Then, we evaluated the expression of anti-microbial peptides (AMPs), including human β defensin (hBD)-1, -2, -3, -4, LL37, and RNase 7. First, we analyzed the mRNA expression in EGFRIs only, or rhEGF and EGFRI co-treated keratinocytes that were pre-treated with LPS for 24 h using real-time qPCR. As a result, LPS induced the expression of hBD-2 up to 24.6-fold and hBD-3 up to 13.4-fold, 24 h after treatment (Figure 5a), but LL37 and RNase7 mRNA expressions were only slightly increased in LPS-treated cells (LL37: 1.7-fold; and RNase 7: 2.0-fold; Figure 5b). On the other hand, in cetuximab-treated cells, the increment of mRNA expression was decreased (Figure 5a,b). Notably, rhEGF increased the decreased AMP mRNA expression in a concentration-dependent manner (Figure 5a,b). Similarly, in gefitinib-treated cells, hBD-2 and hBD-3 mRNA expressions were inhibited; and these were increased by rhEGF treatment in a concentration-dependent manner (Figure 5c). However, mRNA expression of LL37 and RNase 7 was not affected by gefitinib, or gefitinib and rhEGF co-treatment. Next, we could confirm the increased hBD-2 and hBD-3 expressions on the protein level (Figure 5d,e). hBD-2 and hBD-3 protein expressions were inhibited by cetuximab or gefitinib treatment, and those decrements were increased by rhEGF treatment (Figure 5d,e).

### 3.6. Skin Pathologic Change Evaluation of the Patients Who Had EGFR Inhibitor-Related Skin Adverse Events (ERSE)

In addition, in clinical trial results, the immunohistochemistry differences between the ERSE group and the control group were observed. As a result, in the ERSE group, Ki-67 expression (21.2%) was lower than the control group’s Ki-67 expression (40.8%) (*p* = 0.015, Figure 6a,b). Similarly, EGFR presentation range of epidermis was (98.3 vs. 84.6)% in the control group and the ERSE group, respectively (*p* = 0.001, Figure 6c,d). In addition, we could observe (14.2 vs. 8.1)% (*p* = 0.069) of Melan-A immunohistochemistry (Figure 6e,f). On the other hand, IL-17 (*p* = 0.038, Figure 6g,h) and TNF-α (*p* = 0.037, Figure 6i,j) expression intensity in the ERSE group was higher than that of the control group in the dermis.

After the EGF ointment of 5 patients, we could observe the rebuilding of the epidermis and the decrement of infiltrated inflammatory cells in the dermis in the 5 patients’ tissues (Figure 7a,b). We also observed the average values of Ki-67, EGFR, and Melan-A, and the stain levels of IL-17 and TNF-α. As expected, in EGF-ointment-treated patients’ tissues, the average values of Ki-67 (28%), EGFR (94%), and Melan-A (8.2%) were changed (Figure 7c–h). Moreover, all of the patient’s proportion of IL-17 and TNF-α were decreased to low stain level (Figure 7i–l).

## 4. Discussion

In this study, we used two types of EGFR inhibitors, cetuximab and gefitinib, to induce EGFRI-derived skin side effects in vitro and ex vivo, respectively. In addition, we treated cetuximab or gefitinib with rhEGF at the same time, to determine whether EGF could improve the side effects of EGFRIs. To this end, it was necessary to confirm whether EGFR signaling was activated when EGFR inhibitors (cetuximab and gefitinib) were treated simultaneously. So, after simultaneously treating cetuximab and rhEGF, pEGFR expression was simultaneously treated with gefitnib and rhEGF, and then AKT and PI3K phosphorylation related to EGFR signaling was examined. pEGFR expression was examined after simultaneous treating of cetuximab and rhEGF, and AKT/PI3K phosphorylation related to EGFR signaling was identified in gefitinib and rhEGF co-treated cells. As a result, as shown in Figure 1, the expression of pEGFR was increased in the group simultaneously co-treated with cetuximab and rhEGF, compared with the cetuximab alone. In addition, EGFR expression was also increased by rhEGF in a concentration-dependent manner. Furthermore, we compared the binding affinity for human EGFR between cetuximab and rhEGF. As a result, rhEGF binds to hEGFR faster than cetuximab. Our guess is that although cetuximab maintained binding with hEGFR for a longer time than rhEGF, EGFR phosphorylation occurred in cetuximab and rhEGF co-treated cells because of the faster binding rate of rhEGF with hEGFR. Gefitinib is one of the EGFR-TKI inhibitors, and the inhibition of EGFR phosphorylation and phosphorylation of downstream effectors AKT in cells treated with over 0.1 M concentration of gefitinib were observed [25]. Similarly, in this study, gefitinib treatment inhibited the phosphorylation of AKT and PI3K. However, in gefitinib and rhEGF co-treated cells, phosphorylated AKT and PI3K were also observed. We estimate that the lowest concentration of rhEGF (1 ng/mL) in this study exceeds the ability of TKI inhibition of 1 uM gefitinib, but further study is needed.

The skin barrier function is mainly provided by keratinocytes, and it is maintained by a tightly controlled balance between the proliferation and differentiation of keratinocytes [26]. EGFR signaling plays an important role in the final differentiation of keratinocytes by inducing the activation of Transglutaminase (TGase) via Phospholipase C-γ (PLCγ) and Protein kinase C (PKC) [27]. Furthermore, EGF induces keratinocyte proliferation via the Raf-MEK-ERK signaling pathway. In this study, in cetuximab- or gefitinib-treated tissue, low expressions of both ki-67 and filaggrin were observed. However, in rhEGF and cetuximab or gefitinib co-treated tissue, those expressions were high, and the expressions were similar to those of untreated control tissue. These results mean that rhEGF treatment could normalize the proliferation and differentiation of keratinocytes via activating the EGFR signaling pathway. Keratins are the principal structural proteins in the epidermis, and the primary function of keratin is structural and mechanical support. In addition, keratins also modulate the growth, adhesion, migration, and invasion of epithelial cells. So, the dysfunction or mutations of keratin proteins are associated with a remarkable variety of skin disorders, including skin blistering and inflammatory disorders [26]. In normal skin, the epidermis expresses K1 and K10 in suprabasal layers, and K5 and K14 in the basal layer, whereas in inflammatory skin, as in atopic dermatitis and psoriasis, the expression of inflammatory related-keratin, including K6, K16, and K17, increases [28]. Terrinoni et al. reported that the abnormal expression of K10 with respect to control skin was observed in the lesional skin of patients who were under epidermal erythrodermic hyperkeratosis [29]. In this study, K5 expression was inhibited by EGFRIs, and this decrement was induced by rhEGF co-treatment (Figure 2c). The aggregation of K10 in EGFRIs-treated tissues was observed, but in rhEGF- and EGFRIs-treated tissues, the aggregation of K10 was not observed (Figure 2d). These results mean that the interruption of EGFR signaling by EGFRIs also affects keratin expression, and further studies of other inflammatory keratins, such as K6, K16, and K17, in the lesional skin of patients under EGFRI-induced skin toxicity are needed.

EGFR is also implicated in regulating cell–cell contacts. The expression of claudin-3 was reduced in mice deficient in Egfr keratinocytes (*EGFR^dEP^*), similarly to the expression of claudin-1 in human lesional skin [30]. As expected, we also confirmed the reduction of claudin-1, -3, and occludin expression in EGFRI-treated tissues. Meanwhile, there were no dramatic changes of tight junction abnormal in rhEGF- and EGFRI-treated tissues (Figure 3). However, a weak normalization of tight junction expression was observed in rhEGF and EGFRI co-treated tissues. Therefore, these results mean that rhEGF may indirectly affect the formation of tight junctions during normalization of the proliferation and differentiation of keratinocytes.

Our previous studies reported that topical treatment of rhEGF relieved S. aureus-induced inflammation and AD-like skin lesions in Nc/Nga mice, and rhEGF treatment attenuated P. acne-induced inflammatory responses, at least in part, through the modulation of TLR2 signaling in human epidermal keratinocytes [18,20]. In this study, proinflammatory cytokines, including IL-1α, IL-8, and TNF-α expression, were increased by EGFRIs, and those were down-regulated by rhEGF (Figure 4). EGFR signaling in keratinocytes affects the expression of chemokines induced by TNF-α through the ERK1/2 signaling pathway. Activated ERK1/2 has been shown to decrease chemokine mRNA stability and decrease chemokine mRNA expression, suggesting that EGFR-dependent ERK1/2 activity in keratinocytes also participates in homeostatic mechanisms that regulate dermatitis response [31].

The skin is not only a physical barrier protecting from infection, but also an environmental niche hosting a plethora of commensal organisms. Combined with the physical epidermal barrier, antimicrobial peptides (AMPs) are major components of the active innate immune defense against invading microbes in the skin. Several reports revealed that the expression of some kinds of AMPs is under the control of EGFR. *Helicobacter pylori* virulence effector CagA or LPS function via EGFR signaling to either suppress or upregulate the expression of hBD3, respectively [32,33]. The treatment of human epidermal keratinocytes with erlotinib reduced the expression of hBD3, RNase 7, and CAMP. Similarly, in *egfr*-deficient keratinocytes, it was observed that the expression of murin β-defensin 14, the mouse homolog of hBD3, was reduced [34]. In this study, we confirmed that mRNA expression of AMPs, including hBD-2, -3, LL37, and RNase7, was inhibited by EGFRI treatment (Figure 5a–d). Moreover, mRNA expression of AMPs was increased in rhEGF and EGFRI co-treated cells (Figure 5a–d). However, only hBD-2 and -3 protein expressions were affected by EGFRIs and rhEGF (Figure 5e,f). These results mean that rhEGF may be used to improve the innate immunity of the epidermis via inducing the expression not of cathelicidins, but of defensins, especially hBD-2 and -3, through EGFR signaling.

Clinical trial results confirmed that EGFRIs decreased the proliferation and differentiation of keratinocytes. Moreover, EGFRIs induced inflammatory reactions in the skin. As expected, EGF ointment induced the re-epithelization and proliferation of keratinocytes, and reduced infiltrated inflammatory cytokines in the dermis. These results are remarkably similar to the in vitro and ex vivo results in this study. Nevertheless, further studies are needed, because in this study the expression of tight junction and AMPs in lesional skin was not confirmed.

## 5. Conclusions

In conclusion, EGFRIs: (1) inhibit the proliferation and differentiation of keratinocytes, (2) cause the eruption of the keratinocytes’ differentiation process, (3) cause tight junction protein expression abnormalities that play an important role in physical barriers, (4) induce inflammatory factors and play an important role in the innate immunity of the epidermis, inducing the inhibition of AMP expression. EGFRIs cause epidermal physical and immunological barrier abnormalities, resulting in complex skin side effects, such as xerosis and rash. In this study, we revealed that the treatment of rhEGF in human epidermal keratinocytes, 3d-cultured human skin tissue, and patient lesions improved EGFRI-induced skin eruption via normalizing the proliferation and differentiation of keratinocytes, reducing inflammatory cytokines expression and inducing the expressions of hBD-2 and hBD-3. These findings may therefore support the effect of topical rhEGF treatment on the improvement of EGFRI-derived skin side effects.

## Figures and Tables

**Figure 1 cancers-12-03120-f001:**
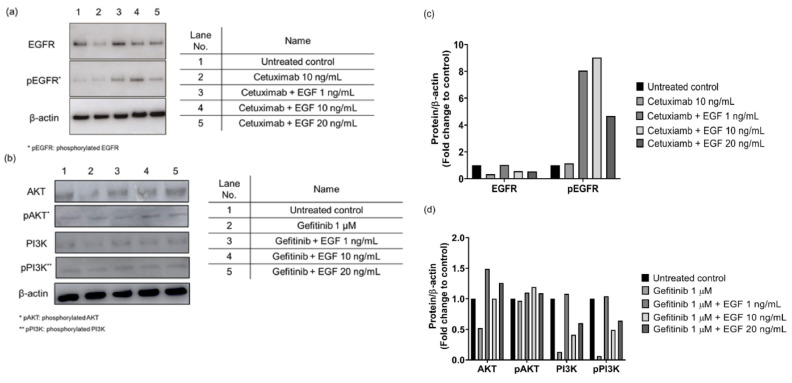
The recombinant human epidermal growth factor (rhEGF) increased the epidermal growth factor receptor inhibitor (EGFR) and EGFR signaling was interrupted by EGFR inhibitors. The expression of the EGFR of human epidermal keratinocytes was decreased by cetuximab treatment. However, EGFR and phosphorylated EGFR expression was induced by rhEGF treatment (**a**). Similarly, in gefitinib-treated keratinocytes, it was observed that phosphorylated AKT and PI3K were slightly decreased. On the other hand, rhEGF and gefitnib co-treatment induced AKT and PI3K phosphorylation (**b**). Representative Western blot images and relative densitometric bar graphs of EGFR and phosphorelated EGFR (**c**) β-actin was used as protein loading control. rhEGF treatment induced pEGFR expression decreased by cetuximab (**c**). Representative Western blot images and relative densitometric bar graphs of AKT, PI3K, phosphorylated AKT, and phosphorylated PI3K (**d**) β-actin was used as protein loading control. Gefirinib treatment inhibited the expression of EGFR signaling related molecules. That decrement was increased by rhEGF (**d**).

**Figure 2 cancers-12-03120-f002:**
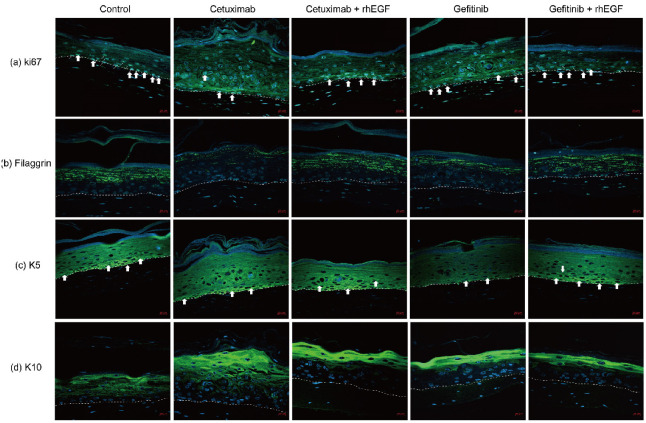
rhEGF normalized proliferation and differentiation of epidermis-erupted EGFRIs. Expression of ki67 in the nucleus of keratinocytes in the epidermal basal, para-basal cell layer (white arrow) (**a**). Expression of filaggrin in upper epidermis (**b**). K5 expression was seen in the epidermal basal layer (white arrow) (**c**). K10 expression was observed in upper epidermis, especially in the stratum corneum (**d**). (Magification, ×20, Scale bar: 20 μM).

**Figure 3 cancers-12-03120-f003:**
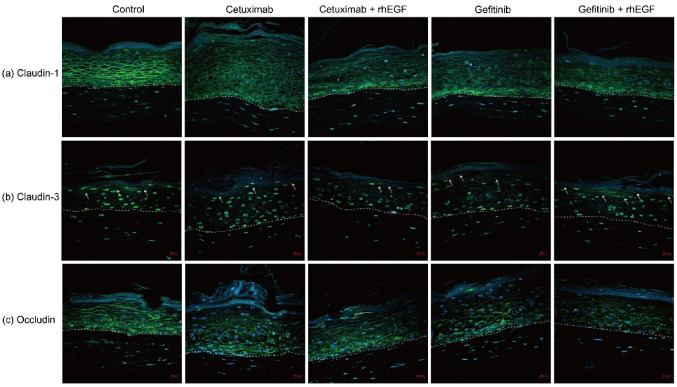
Tight junction expression was erupted by EGFRIs, which was weekly normalized by rhEGF. Claudin-1 expression was seen in epidermal layers, especially the suprabasal layer (**a**). Expression of claudin-3 in stratum corneum and stratum granulosum junction (**b**). Expression of occludin was observed in all epidermal layers (**c**). (Magification, ×20, Scale bar: 20 μM).

**Figure 4 cancers-12-03120-f004:**
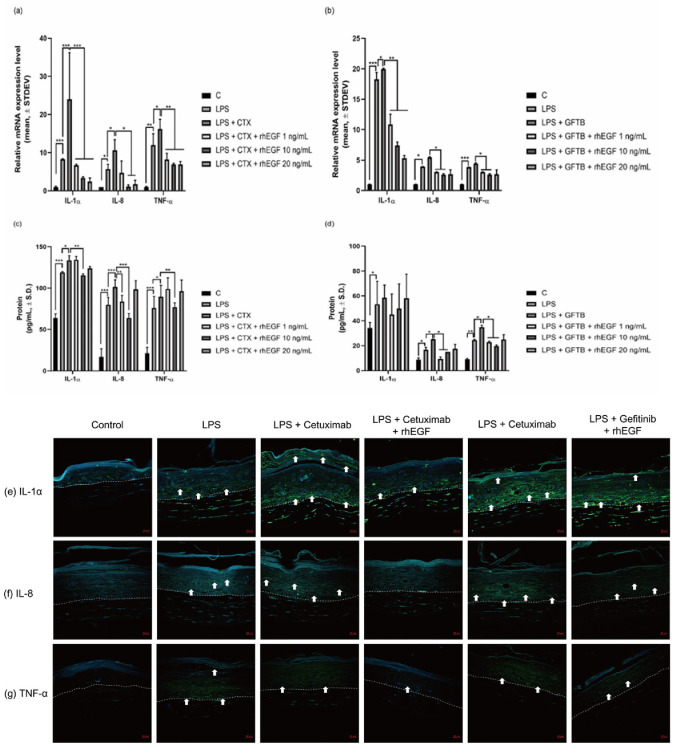
EGFRI-induced inflammatory cytokine expression was inhibited by rhEGF treatment. IL-1α, IL-8, and TNF-α mRNA expression was increased in keratinocytes treated with cetuximab or gefitinib (**a**,**b**). However, that increment was decreased by rhEGF and cetuximab (**a**) or gefitinib (**b**) co-treatment. The protein expression of IL-1α, IL-8, and TNF-α also induced cetuximab (**c**) or gefitinib (**d**) treated keratinocytes (**c**,**d**), and that was inhibited by rhEGF and EGFRI co-treatment (**c**,**d**). IL-1α expression was observed in the epidermal layer (white arrow) (**e**). The expression of IL-8 was slightly observed in the epidermis (white arrow) (**f**). TNF-α expression was seen in the epidermal layer (white arrow) (**g**). Results are presented as mean ± standard deviation (S.D.), representative of three separated experiments. Asterisks indicate statistically significant differences (* *p* < 0.05, ** *p* < 0.01, *** *p* < 0.001). (Magification, ×20, Scale bar: 20 μM).

**Figure 5 cancers-12-03120-f005:**
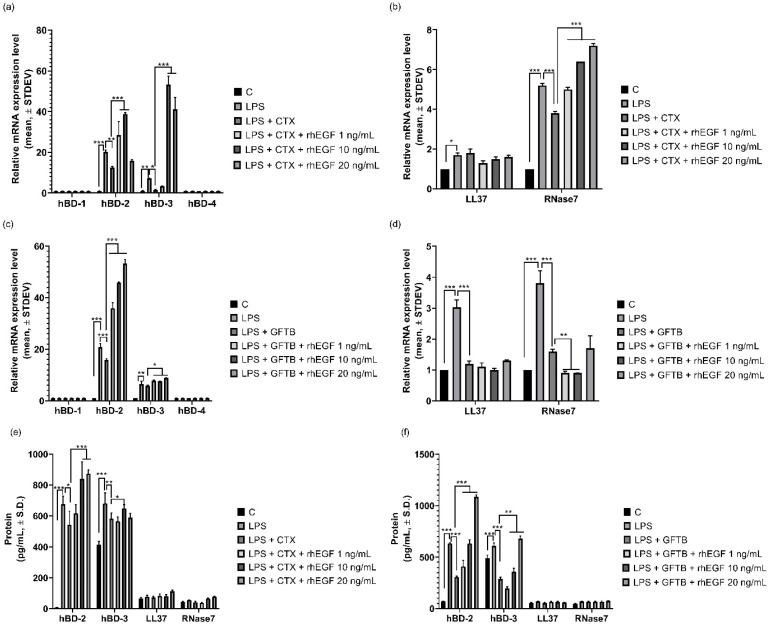
EGFRI-inhibited AMP expression was increased by rhEGF treatment. LPS increased mRNA and protein expression of antimicrobial peptides, including hBD-2, -3, LL37, and RNase 7, in human epidermal keratinocytes (**a**–**f**). It was observed that mRNA and protein expression of hBD-2 and -3 was decreased in LPS- and EGFRI-treated cells. Also, rhEGF increased hBD-2 and -3 mRNA and protein expression respectably (**a**,**c,e,f**). However, in rhEGF- and cetuximab-treated cells, rhEGF only increased RNase 7 but not LL37 mRNA expression (**b**). Unlike the mRNA expression results, rhEGF did not affect the protein expression of LL37 and RNase 7 (**e**). Similarly, in rhEGF and gefitinib treated cells, rhEGF affected only RNase 7 mRNA expression (**d**) but not the protein expression of both LL37 and RNase 7 (**f**). Results are presented as mean ± standard deviation (S.D.), representative of three separate experiments. Asterisks indicate statistically significant differences (* *p* < 0.05, ** *p* < 0.01, *** *p* < 0.001).

**Figure 6 cancers-12-03120-f006:**
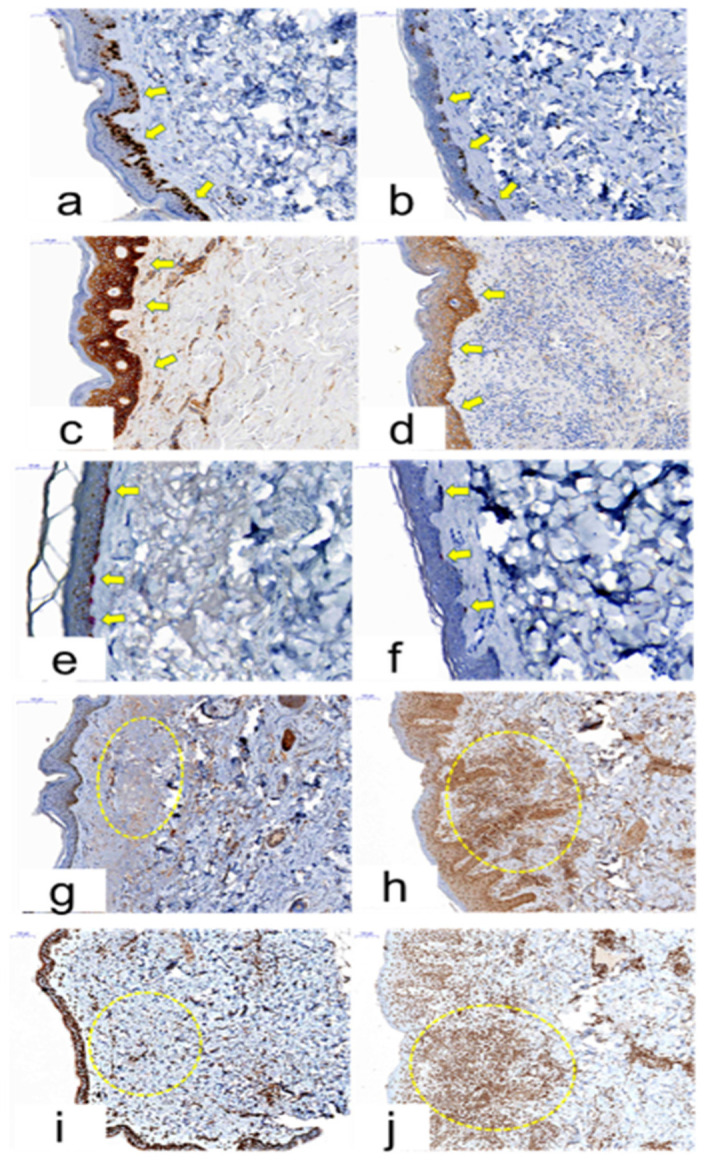
Pathologic comparison between normal control and patient who had EGFRI-related side effects (ERSE). (x 200) Expression of Ki-67 in the nucleus of keratinocytes in the epidermal basal, para-basal cell layer (**a:** control, **b:** ERSE). Expression of EGFR in the membrane of keratinocytes in the epidermal basal cell layer (**c:** control, **d**: ERSE). Cytoplasmic Melan-A expression was seen in the basal melanocytes of the epidermis (**e**: control, **f:** ERSE). Cytoplasmic IL-17 expression in the dermis (**g**: control, **h**: ERSE) and Nuclear with or without cytoplasmic TNF-α expression in the dermis and epidermis (**i:** control, **j:** ERSE).

**Figure 7 cancers-12-03120-f007:**
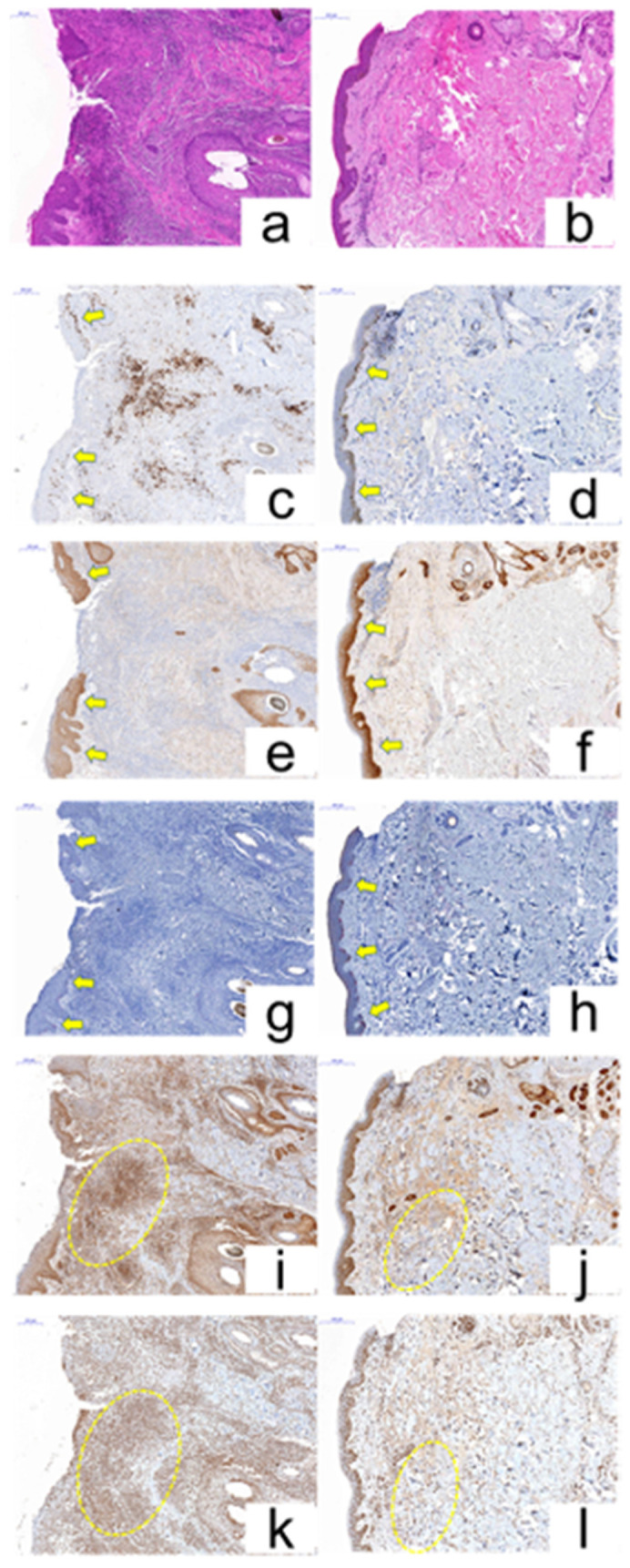
ERSE skin pathologic change pre- and post-EGF ointment treatment. (x 200) H&E stain (**a**: pre-, **b**: post-EGF treatment). The expression of Ki-67 in the nucleus of keratinocytes in the epidermal basal, para-basal cell layer (**c:** pre-, **d:** post-EGF treatment). The expression of EGFR in the membrane of keratinocytes in the epidermal basal cell layer (**e:** pre-, **f:** post-EGF treatment). Cytoplasmic Melan-A expression was seen in the basal melanocytes of the epidermis (**g:** pre-, **h:** post-EGF treatment). Cytoplasmic IL-17 expression in the dermis (**i:** pre-, **j:** post-EGF treatment) and Nuclear with or without cytoplasmic TNF-α expression in the dermis and epidermis (**k:** pre-, **l:** post-EGF treatment).

**Table 1 cancers-12-03120-t001:** rhEGF and cetuximab were compared in a surface plasmon resonance (SPR) analysis for their human EGFR binding, and approximate kinetic parameters are presented.

Ligand	Analyte	Conc.	K_a_ (1/Ms)	K_d_ (1/s)	K_D_ (M)	Rmax	Chi^2^
Human EGFR	rhEGF	1.563, 3.125, 6.25, 12.5, 25, 50, 100, 200 nM	6.07 × 10^5^	7.91 × 10^−4^	1.30 × 10^−9^	11.8	0.497
	Cetuximab	0.012, 0.024, 0.049, 0.098, 0.195, 0.391, 0.781, 1.563, 3.125, 6.25, 12.5, 25, 50, 100, 200 nM	3.54 × 10^5^	1.75 × 10^−4^	0.49 × 10^−9^	292.8	8.11

EGFR: epidermal growth factor receptor, rhEGF: recombinant human epidermal growth factor.

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
