# Peer review of "rhEGF Treatment Improves EGFR Inhibitor-Induced Skin Barrier and Immune Defects"

_cancers, 2020, doi:10.3390/cancers12113120_

Round 1

Reviewer 1 Report

Thanks to dr JM Kim and colleagues for presenting a mechanistic insight into skin toxicities caused by EGFR receptor inhibitor. The authors referred to their previous phase III trial (NCT02284139), which showed EGF ointment to be effective for managing EGFR inhibitor‐related skin adverse events (ERSEs) and improved the quality of life (Kim SY et al. A Randomized Controlled Trial of Epidermal Growth Factor Ointment for Treating Epidermal Growth Factor Receptor Inhibitor‐Induced Skin Toxicities. Oncologist 2020, 25, e186-e193). The skin toxicity is actually reported in decreasing extent under the current standard of care with Osimertinib, but still present. All grade rash and acne were observed in 58% in Osimertinib arm (grade 3 in 1%) and 78% in standard EGFR-TKIs arm (grade 3 in 7%) (Soria J-Ch et al. Osimertinib in Untreated EGFR-Mutated Advanced Non–Small-Cell Lung Cancer. January 11, 2018. N Engl J Med 2018; 378:113-125 DOI: 10.1056/NEJMoa1713137).
The study is based on well-designed experiments and showed that expression of pEGFR was increased in the group simultaneously co-treated with Cetuximab and rhEGF, compared with the Cetuximab alone. In addition, EGFR expression was also increased by rhEGF in a concentration-dependent manner. The authors interpretate that rhEGF binds to hEGFR faster than does Cetuximab and EGFR phosphorylation occurred in Cetuximab and rhEGF co-treated cells. In Gefitinib and rhEGF co-treated cells, phosphorylated AKT and PI3K were also observed. Figure 2 present that proliferation and differentiation of epidermis caused by Cetuximab and Gefitinib can be reversed by rhEGF. Figure 3 perfectly illustrate changes in the status of the measured inflammatory parameters in response to Cetuximab, Cetuximab + rhEGFR, Gefitinib, Gefitinib + rhEGFR compared with control. Observations presented in Figure explain that 4 proinflammatory cytokines, including IL-1α, IL-8, and TNF-α expression, were increased by EGFRIs, and those were down-regulated by rhEGF.  Figure 6 and 8 visualize different aspects of the condition of the skin affected by ERSEs comparing with normal control and after EGF ointment treatment. An interesting observation regarding the immunity of the skin was also detected regarding overexpression of defensins through EGFR signaling. Furthermore, skin toxicities is one of the factors having significant impact on quality of life as presented in recently published study in patients treated with first and second generation EGFR-TKI, where majority of patients (93,2%) experienced this adverse event (L-Ch Tseng et al. Effects of tyrosine kinase inhibitor therapy on skin toxicity and skin-related quality of life in patients with lung cancer: An observational study. Medicine (Baltimore) 2020 Jun 5;99(23):e20510.). As skin symptoms correlated significantly with poor quality of life (r=0.50, P<.001), this problem still represent an unmet need.  
In conclusion the work is a good contribution to understanding how rhEGF works, which can be a good alternative for the treatment of skin toxicities instead of antibiotics and steroids only.

I have one question:
Based on your current research, do you expect any problems in EGFR-mutated NSCLC patients with skin metastases treated with EGFR-TKI, where both ERSRs occur and you use rhEGF to treat them?

Besides:  

Line 69 - ERSEs as used the first time, therefore please write the entire acronym in parenthesis.

Author Response

  1. Based on your current research, do you expect any problems in EGFR-mutated NSCLC patients with skin metastases treated with EGFR-TKI, where both ERSRs occur and you use rhEGF to treat them?

=> Thank you for your interesting question. Generally, EGF ointment does not affect the systemic effects of EGFR-TKI as it is not absorbed into the blood vessels due to its large molecular weight. Skin metastasis of lung cancer is very rare and is known to be less than 1%. If the patients have lung cancer with skin metastasis and they are using EGFR-TKI, theoretically, applying EGF ointment to metastatic site and ERSE sites can interfere with the operation of EGFR-TKI, so I don’t want to recommend use it.

  1. Line 69 - ERSEs as used the first time, therefore please write the entire acronym in parenthesis.

=> Thank you. In response to Reviewer 1’s comment, the full abbreviation in parentheses for ERSEs was added on line 73. EGFRIs related skin adverse effects (ERSE)

Reviewer 2 Report

Comments file attached 

Author Response

Reviewer 2

  1. The findings should be incorporated with in abstract section.

=> Thank you for your precise indication. I revised abstract as below

Based on Reviewer 2's comment, I revised and rewritten the abstract.

1) The sentences on line 36, 37, 38, 39 were deleted.

2) The words “As a result” were added on line 39.

3) “In addition, rhEGF bound to EGFR faster than did cetuximab, however cetuximab bound to EGFR stronger than did rhEGF” was added on line 41.

4) That sentence on line 45 was modified as below:

   Before: Pro-inflammatory cytokines, including IL-1α, IL-8 and TNF-α expression were increased by      EGFRIs, and down-regulated by rhEGF.

   After: Expression of IL-1α, IL-8, and TNF-α was increased by EGFRIs, and down-regulated by rhEGF.

5) “Also, hBD-2 and hBD-3 protein expressions were inhibited by cetuximab or gefitinib treatment, and those decrements were increased by rhEGF treatment” was added on line 46.

6) On line 52, “In conclusion” was added.

  1. In the abstract, there are multiple errors, likewise, line 38 after IHC stain the author wrote “or” which should be “and” and on the same line the next following sentence is having no relevance with the contents also no need of the methodology of the assay in the abstract session. It would be better to rephrase the abstract by highlighting the experimental findings.

=> Thank you for your kind mention. Based on Reviewer #2 No 1 questions about abstract revision, we revised abstract. Please check revised abstract.

  1. Material and methods section, 2.2. cell culture section, the author needs to describe the selected does of LPS using for the induction for the inflammation. Also please provide the details about the LPS induced toxicity.

=> Thank you for your great comment.

1) In response to Reviewer 2's comment, dose of LPS was described on line 111 and 112 (100 ng/mL of LPS). Also, that dose of LPS using for the induction for the inflammation was selected by references. So, “The concentration of LPS was decided by several references” was added on line 111 and also references were added on the same line.

2) In this study, LPS was treated to keratinocytes for inflammation response induction. So, we confirmed expression of inflammatory cytokines and AMPs in keratinocytes after LPS treatment. As a result, 100 ng/mL LPS induced mRNA and protein expression of inflammatory cytokines including IL-1α, IL-8, and TNF-α and AMP including hBD-2, hBD-3, LL37, and Rnase 7. And these results were showed at Figure 4 and 5. Since the purpose of LPS treatment was to induce an inflammatory reaction, we did not confirm the other toxicity of the inflammatory reaction.

  1. Results: Figure 1b western blot results are not clear and for such an important experiment author needs to provide clear version of the expression of the EGFR signaling entities.

=> Thank you for your comment. The density of the Western blot band was measured and corrected with β-actin, and then the density of each protein compared to the control was compared and drawn as a graph, and the result was modified and reflected in Figure 1 (c) and (d).

  1. Does the ligand binding affinity of the analytes is concentration dependent or independent, please explain?

=> Thank you for your question. It is concentration dependent. In order to measure the binding affinity of rhEGF and cetuximab to human EGFR, a certain amount of human EGFR was attached to the chip, and then various concentrations of ligand were flowed through the chip to obtain the binding rate constant and dissociation rate constant. At this time, there is a difference in the values ​​of the binding rate constant and the dissociation rate constant depending on the concentration of the flowing ligand. This is because the higher the concentration of Ligand, the greater the number of antibodies that bind to the ligand. Therefore, in this study, kinetics affinity was analyzed by confirming the reaction at concentrations of 5 or more for each ligand.

  1. Conclusion should be separate section.

=> Thank you. As your recommendation, “5. Conclusion” was added on line 485. I separated the paragraph of discussion which was written.

Reviewer 3 Report

I would suggest adding the results of rhEGF treatment alone as control of the experiments. It would be interesting to see how much the expression of EGFR, AKT etc. is effected by rhEGF in the absence of EGFR inhibitors. I would also suggest quantifying the expression levels detected by the western blots (Figure 1.) by densitometry or other methods and present the values with bar diagrams. It takes a lot of effort to see the difference especially in the case of PI3K. In the case of the IHC results of the skin grafts, little arrows or circles could also help to notice the differences especially in the case of Ki67 and k5. Also, Figure 4 IHC pictures could have arrows for better understanding. Figure 6 for example is very clear.

Author Response

Reviewer 3

1) I would suggest adding the results of rhEGF treatment alone as control of the experiments.

It would be interesting to see how much the expression of EGFR, AKT etc. is effected by rhEGF in the absence of EGFR inhibitors.

=> Thank you for your interesting suggestion. But, unfortunately, we didn't experiment of rhEGF treatment alone as control. However, there are several references. According to Veronica Dudu et al., (Cell Mol Bioeng. 2012 Dec; 5(4): 502–413.), EGF-induced receptor phosphorylation triggered the downstream activation of phosphoinositide-3 kinase (PI3K)/Akt pathway, while its downstream activation was inhibited by Tarceva (an EGF-R inhibitor), and Wortmannin (a PI3K inhibitor). Moreover, it was reported that EGFR mRNA expression level was elevated in EGF treated human KB carcinoma cells (Adrian J. L. Clark et al., Proc. Natl. Acad. Sci. USA 1985, 82, 8374–8378.)

2) I would also suggest quantifying the expression levels detected by the western blots (Figure 1.) by densitometry or other methods and present the values with bar diagrams.

=> Thank you for your recommendation. The density of the Western blot band was measured and corrected with β-actin, and then the density of each protein compared to the control was compared and drawn as a graph, and the result was modified and reflected in Figure 1 (c) and (d).

3) It takes a lot of effort to see the difference especially in the case of PI3K. In the case of the IHC results of the skin grafts, little arrows or circles could also help to notice the differences especially in the case of Ki67 and k5. Also, Figure 4 IHC pictures could have arrows for better understanding. Figure 6 for example is very clear.

=> Thank you for your comments. Figure 2 (a) ki67 and (c) K5 are displayed with white arrows to make it easier to check the expression of each protein in the tissue. In the results of Figure 4 (e) to (g), white arrows were added to clearly confirm the part of each cytokines expressed on the tissue.
